# The Cell Biology of Heterochromatin

**DOI:** 10.3390/cells11071247

**Published:** 2022-04-06

**Authors:** Brandt Warecki, William Sullivan

**Affiliations:** Department of Molecular, Cellular, and Developmental Biology, University of California, Santa Cruz, CA 95064, USA

**Keywords:** heterochromatin, centromere, chromosome segregation, nuclear envelope reassembly

## Abstract

A conserved feature of virtually all higher eukaryotes is that the centromeres are embedded in heterochromatin. Here we provide evidence that this tight association between pericentric heterochromatin and the centromere is essential for proper metaphase exit and progression into telophase. Analysis of chromosome rearrangements that separate pericentric heterochromatin and centromeres indicates that they must remain associated in order to balance Cohesin/DNA catenation-based binding forces and centromere-based pulling forces during the metaphase–anaphase transition. In addition, a centromere embedded in heterochromatin facilitates nuclear envelope assembly around the entire complement of segregating chromosomes. Because the nuclear envelope initially forms on pericentric heterochromatin, nuclear envelope formation proceeds from the pole, thus providing time for incorporation of lagging and trailing chromosome arms into the newly formed nucleus. Additional analysis of noncanonical mitoses provides further insights into the functional significance of the tight association between heterochromatin and centromeres.

## 1. Introduction

Since its discovery in 1928 [1], much progress has been made regarding the structure and function of heterochromatin. Constitutive heterochromatin consists primarily of late replicating, highly repeated tandem sequences that are concentrated at the centromeres and telomeres [2]. Because of a lack of protein coding genes and the fact that large regions of heterochromatin can be removed without obvious functional consequences, heterochromatin was considered genetically inert and, perhaps unfairly, relegated to the category of junk or selfish DNA. Molecular, genetic, and cellular studies over the past two decades have dramatically changed this view, revealing key functions of these heterochromatic regions. These include protecting telomeres, suppression of transposon activity, and facilitating repair of repeat sequences [3].

A conserved feature of heterochromatin and clue to its function is that the centromeres of virtually all higher eukaryotes are embedded in extensive arrays of tandemly repeated heterochromatin. For example, the recent complete sequencing of the human 155 Mb X chromosome, revealed 3.1 Mb of highly repeated sequence encompassing the centromere [4]. Known as pericentric heterochromatin, the close association with centromere strongly suggests that each may influence the activity of the other [5]. Here, we focus on recent and past cellular studies demonstrating that the tight association between pericentric heterochromatin and centromeres is essential for properly timing chromosome separation and nuclear envelope assembly.

## 2. Pericentric Heterochromatin and Sister Chromatid Separation

The pericentric heterochromatin/centromere complex plays an essential role in sister chromatid pairing. Cohesin is the molecular glue that maintains pairing of sister chromatids. The Cohesin complex consists of three proteins (Smc1, Smc3, and Kleisin (Rad21)) forming a ring that encircles sister chromatids. Cohesin is loaded onto the chromosomes during S-phase and removed in two temporally and spatially distinct steps. In the first step occurring during prophase, the bulk of Cohesin is removed from the arms but remains on the pericentric heterochromatin [6]. Known as the prophase pathway, Cohesin removal from the chromosome arms is driven by the kinases Plk1 and Aurora B and the Wapl protein. Cohesin remains at the pericentric heterochromatin, and the sister chromatids remain paired specifically in the pericentric regions. This readily explains the classic “X” pattern of sister chromatids arrested in metaphase. This phenomenon is known as the prophase protection pathway. This is mediated by Shugoshin complexed with the phosphatase 2A (PP2a) which specifically accumulates at the pericentric heterochromatin through a Bub1- and Aurora A kinase-dependent mechanisms [7,8]. During the metaphase–anaphase transition, APC activation of Separase cleaves Kleisin/Rad21 releasing the paired sisters from the Cohesin linkage [9]. Because the chromosomes have been under constant tension by microtubules, the sister chromatids immediately segregate to opposite poles upon Kleisin cleavage to opposite poles (Figure 1A).

Left unresolved is whether the centromere, the pericentric heterochromatin, or both provides the signals preventing Cohesin removal. In a related issue, it is unclear whether processes in addition to the Shugosin-mediated protection maintain Cohesin at the centromeric regions. To address these issues, we took advantage of chromosome rearrangements in which large blocks of heterochromatin are embedded in euchromatin and physically distant from a centromere. In one chromosome rearrangement that proved particularly informative, known as the compound chromosome two, both homologs have been fused such that they share a common centromere [10,11]. This results in a unique engineered chromosome with arms twice the normal length organized as follows: 2R-2L-centromere-2L-2R (Figure 1B). The right and left arms are connected by Y heterochromatin. Separated from the centromere, the displaced heterochromatin provides a means of determining the role of heterochromatin in Cohesin dynamics independently of the centromere.

This rearrangement and others revealed that the maintenance of Cohesin association with pericentric heterochromatin into late metaphase is independent of its proximity to the centromere [12]. During late metaphase after Cohesin has been removed from the euchromatic arms, in addition to the constriction at the centromeres maintaining sister pairing, additional constrictions are observed at the sites of ectopic heterochromatin. In addition, analyses using FISH and heterochromatic specific markers (Histone H3K9me2) reveal that these constrictions correspond to the displaced heterochromatin and lack CENP-A/CID, the *Drosophila* CENPA histone H3 centromere-specific variant. Significantly, Cohesin is elevated at these ectopic heterochromatic sites.

Based on analysis of Cohesin dynamics at the pericentric heterochromatin in wild-type chromosomes, an obvious explanation for these results is that Cohesin is preferentially removed from the euchromatin regions during prophase resulting in Cohesin enrichment at both the normal and displaced ectopic heterochromatin. However, results from experiments were not in accord with this idea. Shugosin, the prophase protective protein, could not be detected at the ectopic heterochromatic sites [12]. In addition, enrichment of Cohesin at the ectopic heterochromatic sites was even observed in Wapl mutants, the gene responsible for removing Cohesin during prophase.

These findings suggest an alternative model: Cohesin is preferentially loaded at both the normal and ectopic heterochromatic blocks. Localization studies of the *Drosophila* Cohesin loading factor, Nipped B, revealed that it is enriched at the pericentric heterochromatin during S-phase but is no longer detectable by metaphase. Significantly in the compound chromosome two lines, in addition to the pericentric heterochromatin, Nipped B is enriched at the two ectopic heterochromatin sites [12]. Thus, in *Drosophila*, the enrichment of Cohesin at the ectopic heterochromatin is in part due to Nipped B loading at these sites during S-phase. Another possibility is that signals emanating from the centromere promote the timely removal of Cohesin and decatenation of the pericentric heterochromatin.

### 2.1. Delayed Cohesin Removal at Ectopic Heterochromatic Results in Delayed Sister Separation at These Sites

While the dynamics of Cohesin loading is similar between pericentric and ectopic heterochromatin, the dynamics of Cohesin removal is delayed at the ectopic heterochromatin. Live imaging Cohesin removal from the compound chromosome during the metaphase–anaphase transition in *Drosophila* neuroblasts revealed that Cohesin removal occurs but is significantly slower than at the normal pericentric heterochromatin sites [12]. During anaphase, extensive stretching of the chromatin is observed at these ectopic sites. Stretching appears to be a universal feature of ectopic heterochromatin as all rearrangements with displaced heterochromatin exhibit this phenomenon. These studies also revealed a strong correlation between constriction and stretching: a higher degree of constriction correlates with greater stretching. It is likely that high levels of constriction are due to increased levels and delays in Cohesin removal. Significantly, a greater distance of ectopic heterochromatin from the centromere results in more stretching. Stretching also occurs in the chromosome region between the centromere and the constricted ectopic heterochromatin, and this region remains stretched well after Cohesin has been removed. These observations are in accord with previous studies demonstrating that timely removal of Cohesin at the heterochromatic sites is necessary for clean separation of the sister chromatids [13].

In addition to Cohesin-mediated pairing, sister chromatids remain joined through intertwining of the sister DNA molecules. An inevitable outcome of DNA replication is a looping or catenation of the newly replicated sister DNA molecules which must be resolved through the action of topoisomerase and other mechanisms [14]. In fact, these catenations are sufficient to maintain sister pairing even in the absence of Cohesin. Cohesin removal is required to resolve catenations [15,16]. Thus, delayed removal of Cohesin at pericentric heterochromatin results in a shortened time window for decatenation compared to that of the chromosome arms.

Given that decatenation is dependent on Cohesin removal, it is puzzling why the stretching and delayed separation is observed at the ectopic heterochromatic sites but not at the normal pericentric heterochromatin regions. Since both exhibit delayed Cohesin removal, one would expect both to behave similarly with respect to sister segregation dynamics. The explanation likely lies in the fact that microtubule generated opposing forces at sister kinetochores provides tension facilitating decatenation of the pericentric heterochromatin [15,17,18]. Chromatin regions further from the centromere experience reduced spindle forces. Thus, heterochromatin displaced from its centromere is under much less tension than pericentric heterochromatin and consequently undergoes delayed decatenation, resulting in stretched chromatin at these regions (Figure 1B). This model is supported by the observation that the greater the distance the heterochromatin is from the centromere, the greater the degree of stretching [12]. As described above, it is also conceivable that centromere-derived signals promote timely decatenation of neighboring heterochromatin.

Heterochromatin islands on the maize chromosomes, known as knobs, provide another example of heterochromatin not associated with a centromere [17]. Significantly, sister chromatid separation is delayed at the knob regions resulting in bridges and lagging chromatin [18]. However, in the presence of the abnormal chromosome Ab10, the knobs acquire neocentromeric activity [19]. Recent studies demonstrate that the Ab10 homolog encodes two divergent members of the minus-end microtubule motor protein kinesin-14, Kindr and Trkin [20,21]. Each localizes to a specific class of knobs facilitating their lateral association of with microtubules. The result is precocious segregation of the neocentromeric knobs, ultimately resulting in a meiotic drive [22]. It would be interesting to determine the dynamics and timing of Cohesin loading and removal at both inactive and neocentromeric knobs. Based on the studies of displaced centric heterochromatin in *Drosophila* [12], it is likely that the inactive and neocentromeric knobs would exhibit delayed and precocious Cohesin removal, respectively.

### 2.2. Pericentric Heterochromatin and Centromeres Balance Binding and Pulling Forces on the Sister Chromatids

The excess loading and delayed removal of Cohesin from the pericentric heterochromatin is a highly conserved feature of eukaryotes. Support for this comes from the observation that images of cells arrested in numerous species exhibit the tell-tale “X” configuration. The two-stage removal of Cohesin, first from the arms and then the pericentric heterochromatin, ensures that sister chromatids remain paired until anaphase entry. In addition, the initial removal from the arms restricts Cohesin to the much smaller pericentric heterochromatin regions upon anaphase entry and facilitates the simultaneous removal from all the paired sisters upon anaphase entry. Countering these advantages is the fact that the delayed removal of Cohesin leaves insufficient time to resolve sister DNA catenations at these heterochromatic regions. The solution is through the intimate association of the heterochromatin with the centromere. The centromere connected to microtubules provides opposing pulling forces that efficiently resolve the heterochromatic catenations.

Insight into the pericentric heterochromatin/centromere relationship is not only derived from the studies of displaced heterochromatin described above, but also from studies of displaced centromeres no longer associated with heterochromatin. Without the countering binding forces of the pericentric heterochromatin, the displaced centromeres might be expected to exhibit premature segregation in these regions (Figure 1C). Support for this idea comes from studies of neocentromeres that have formed spontaneously in cell lines in which the native centromere has been removed via CRISPR [23]. The newly formed centromeres exhibit reduced sister chromatin Cohesin (Figure 1C). This may in part be due to a lack of surrounding heterochromatin maintaining Cohesin through DNA catenations. Similarly, neocentromeres in *Drosophila* and humans are also associated with chromosome segregation defects and cancer [24,25,26]. While these defects are the result of two active centromeres on a single chromosome, a lack of sister Cohesin may also be a contributing factor.

## 3. Heterochromatin and Nuclear Envelope Assembly

Throughout interphase, heterochromatin is often localized at the nuclear periphery where it forms both direct and indirect connections with the nuclear envelope [27]. For example, heterochromatin protein 1 (HP1), which binds methylated histone H3K9 heterochromatin [28] also binds to the integral inner nuclear membrane protein lamin B receptor [29,30]. Heterochromatin–nuclear envelope interactions play critical roles in heterochromatin function. For instance, mutants of the chromatin-binding inner nuclear membrane protein Lem2 result in decreased transcriptional repression of pericentric heterochromatin [31]. Therefore, in cells where the nuclear envelope is broken down during mitosis and rebuilt in telophase, re-establishing the heterochromatin–nuclear envelope connections is of the utmost importance.

### 3.1. Initiation of Nuclear Envelope Assembly Often Occurs in Areas of High Heterochromatin Abundance

Some of the first events in nuclear envelope formation involve the recruitment of certain nuclear envelope components to areas with a high abundance of heterochromatin. Following centromere-driven chromosome segregation, pericentric heterochromatin is grouped on the leading, pole-facing edges of daughter nuclei [32]. This can be detected through FISH probes targeting heterochromatin [12] or through the visualization of HP1 on the leading edges of segregated chromosomes [33,34]. Broadly, this pole-facing edge corresponds to the location where certain members of the nuclear envelope (e.g., lamin B, nuclear pore complexes) begin to assemble [34,35,36]. From this initial point, assembly continues around the rest of the nucleus and is completed on the midzone-facing edge of the nucleus (Figure 2A).

Given the position of pericentric heterochromatin during the anaphase–telophase transition, it is likely that initial recruitment of some nuclear envelope components to the pole-facing edge of daughter nuclei is specified by the heterochromatin localized there. Such a hypothesis is supported by multiple lines of indirect evidence. (1) In mammalian cells, heterochromatin–nuclear envelope interactions are reestablished during the formation of the nascent nuclear envelope [37,38,39]. A key step in this process is HP1-mediated recruitment of a nuclear lamina-associated protein, PRR14, to methylated H3K9 chromatin in anaphase [38]. This is followed by lamin assembly on methylated H3K9 heterochromatin during telophase [39]. (2) In an in vitro assay modeling nuclear envelope formation, the introduction of a recombinant HP1 fragment blocks lamin assembly onto chromatin [40]. This effect is reminiscent of a dominant negative mutation. (3) Mild depletion of HP1 by RNAi in *Drosophila* results in decreased initiation of lamin assembly on only the poleward-face of daughter nuclei [34]. (4) The nuclear pore protein ELYS/MEL-28 binds chromatin [41,42], potentially with a preference for AT-rich regions [41], which are common in centromeric DNA [43]. Although AT-binding may not be necessary for its chromatin localization [42], ELYS/MEL-28 localizes with kinetochores throughout mitosis where it subsequently recruits assembling nuclear pore complexes to telophase chromatin in *C. elegans* and human cells [44,45,46].

While these studies suggest that heterochromatin and heterochromatin-associated proteins influence nuclear envelope assembly dynamics, it is worth noting that not every component of the nuclear envelope initially localizes to the pole-facing edges of daughter nuclei. For example, the nuclear envelope component LAP2α first localizes to the midzone-facing edges of daughter nuclei [35]. Clearly, separate mechanisms govern the recruitment of different nuclear envelope components to nascent daughter nuclei. Additionally, some of the nuclear envelope components that bind heterochromatin elements can also bind DNA directly [47]. In summary, heterochromatin may enhance recruitment of nuclear envelope components to telophase nuclei but may not be necessary for nuclear envelope formation.

### 3.2. Pole-to-Midzone Nuclear Envelope Assembly Provides Additional Time for Segregating Chromosomes to Form a Single Nucleus

One effect of initiating nuclear envelope assembly on the pole-facing edge of chromosomes is to quickly reestablish the heterochromatin–nuclear envelope contacts necessary for proper gene regulation [37,38,39]. Another effect is to provide more time for lagging chromosomes or trailing chromosome arms to rejoin segregated nuclei before the nuclear envelope is fully built. Failure to integrate lagging chromosomes or chromosome arms into nuclei can result in aneuploidy and subsequent lethality. The benefit of initiating nuclear envelope assembly on the pole-facing edge of nuclei is illustrated in *Drosophila*, which have the remarkable ability to retain lagging and broken chromosomes with high efficiency [48,49,50]. *Drosophila* neuroblasts initiate lamin and nuclear pore complex assembly on the pole-facing edges of daughter nuclei [34,36]. In these cells, lagging chromosome fragments that segregate soon after anaphase onset rejoin nuclei prior to lamin assembly on the midzone-faces of the nuclei [51]. However, chromosome fragments that segregate well after anaphase onset, when lamin is already assembling on the midzone-face of nuclei, require the formation of a specialized channel in the nascent nuclear envelope on the midzone-face of the nucleus. Without channels, the fragments are excluded as micronuclei [51]. This difference between late- and very late-segregating chromosome fragments highlights that a pole-to-midzone pattern of nuclear envelope assembly aids in maintaining euploidy (Figure 2B).

Additional mechanisms promote initiation of nuclear envelope assembly from the pole by suppressing assembly at other locations. For example, work in cultured *Drosophila* cells demonstrated that lamin and nuclear pore complexes that regularly assemble on segregated daughter nuclei at the poles do not assemble on chromosomes lagging at the midzone [49]. These experiments led to the proposition of an anaphase–telophase checkpoint that prevents nuclear envelope assembly on chromosomes until after they segregate [49,52], although the nature of this checkpoint is debated [53]. Blockage of nuclear envelope assembly on lagging chromosomes occurs even when the lagging chromosomes contain a large amount of heterochromatin, demonstrating that other factors can dominate over heterochromatin when guiding nuclear envelope assembly [34].

### 3.3. Variations of Mitosis Provide Additional Insights into the Functions of Heterochromatin–Nuclear Envelope Interactions

Further insight into the functional relationships between heterochromatin and the nuclear envelope can be gained by studying organisms that undergo variations of mitosis. In *S. pombe*, which divides without breaking down the nuclear envelope, inner nuclear membrane proteins tether centromeric heterochromatin to the spindle pole body (the yeast equivalent of the centrosome) embedded in the nuclear envelope. Mutants for these inner nuclear membrane proteins result in spindle formation defects, growth defects, and abnormal nuclear envelope structures after division [54,55]. These results suggest that even if heterochromatin–nuclear envelope interactions are not regulating post-mitotic nuclear envelope assembly, they are still vital for other core mitotic events.

Therefore, careful exploration of heterochromatin–nuclear envelope dynamics in organisms undergoing other variations of mitosis is clearly warranted. Further study of nuclear envelope assembly in (1) *C. elegans* and other organisms with heterochromatic elements along the entire length of their chromosomes [56,57,58], or in (2) organisms that eliminate large portions of their heterochromatin early in development [59,60] may be particularly worthwhile.

## Figures and Tables

**Figure 1 cells-11-01247-f001:**
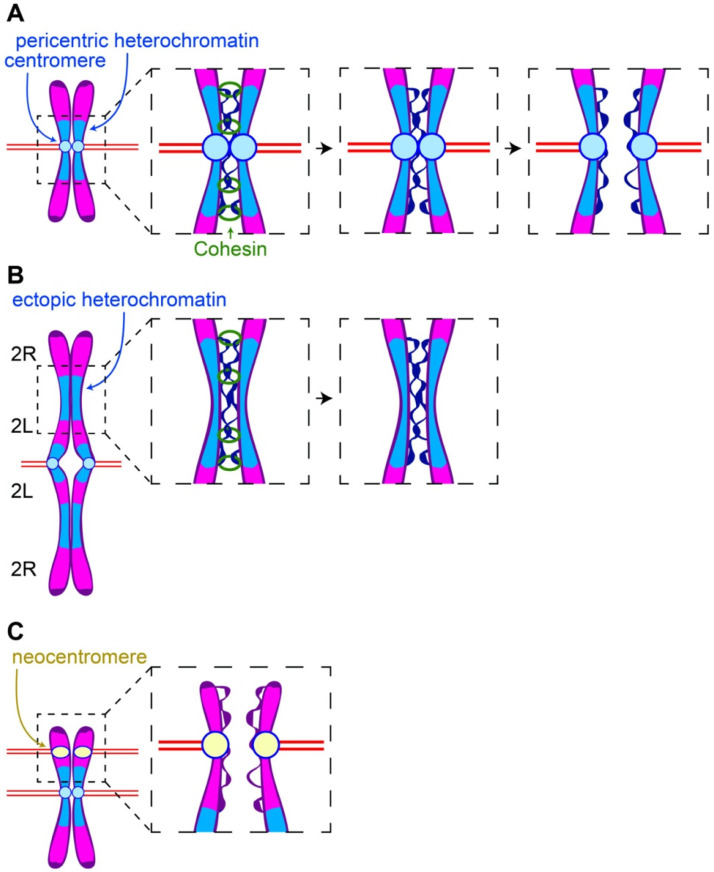
Pericentric heterochromatin and centromeres balance binding and pulling forces on the sister chromatids. (**A**) Once Cohesin is removed, sister chromatids remain bound through DNA catenations at the pericentric heterochromatin. Microtubules attached to opposing kinetochores provide the force required for decatenation and separation of sister chromatids. (**B**) The microtubule pulling force is reduced in the regions in which large blocks of heterochromatin are displaced from their centromeres. Consequently, resolution of sister DNA catenations in the displaced heterochromatin is delayed and results in delayed sister chromatin separation in these regions. (**C**) Segregation defects associated with neocentromeres may be due to the fact that they are no longer associated with heterochromatin that would oppose microtubule pulling forces.

**Figure 2 cells-11-01247-f002:**
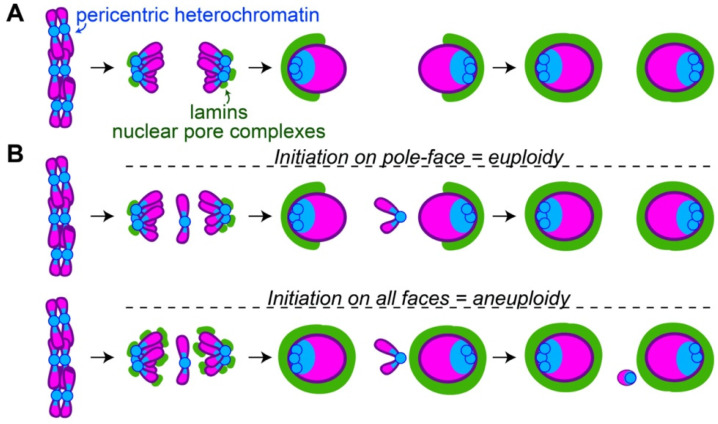
The pole-to-midzone pattern of nuclear envelope assembly on daughter nuclei allows retention of lagging chromosomes or trailing chromosome arms (**A**) Following chromosome segregation, the pericentric heterochromatin (blue) is grouped at pole-facing edges of daughter nuclei. This location corresponds to the site of initial lamin B and nuclear pore complex (green) assembly. Assembly continues from the pole-faces to the midzone-faces of daughter nuclei. (**B**) The pole-to-midzone pattern of nuclear envelope assembly on the daughter nucleus provides extra time for a lagging chromosome to rejoin. In contrast, should nuclear envelope reassembly begin on all faces of the daughter nucleus simultaneously, the nuclear envelope on the midzone-face of the nucleus would form a barrier, excluding the lagging chromosome as a micronucleus.

## Data Availability

Not applicable.

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
