# Peer review of "The Cell Biology of Heterochromatin"

_cells, 2022, doi:10.3390/cells11071247_

Round 1

Reviewer 1 Report

see uploaded pdf

Reviewer 2 Report

This manuscript reviews data on centric heterochromatin in chromosome cohesion and nuclear envelope formation.  Most of the primary papers cited are several years old, and it also cites quite a large number of other reviews (a review of reviews).  I am not really clear on what is new here that compels a new review.   Additionally, while the authors particularly note human and Drosophila data, they (appropriately) cite yeast work, yet do not really integrate the unique features of yeast centric heterochromatin into their discussion.  The absence of typical heterochromatin in S. cerevisae point centromeres and typically eukaryotic organization of the fission yeast highlight some of the points under discussion. 

Reviewer 3 Report

This manuscript is an interesting review for roles of the centric heterochromatin distinguished from those of the centromere during chromosome segregation. The review also correlates the centric heterochromatin with nuclear envelope reformation. I have no serious concerns, but have only some minor comments.

Specific comments

  1. Line 22, “since its discovery in 1928”: From a historical aspect, it would be quite attractive for readers to provide the reference for the first discovery of heterochromatin in 1928.
  2. Line 69: “Wapl-mediated Cohesin removal” appears without any explanation. Some explanation is needed with citation of appropriate references.
  3. Figure 1B: Figure 1B shows two chromosomes fused together at the telomere, but is not explained in the legend or in the text. Figure 1B is first explained much later in line 141, but the corresponding statements only talk about “heterochromatin displaced from its centromere”, but does not describe about the shape of chromosome.
  4. Line 77: Related to the above comment, the “2R-2L-centromere-2L-2R” chromosome does not seem to be the one shown in Figure 1B as the shape of chromosome is different. If this is correct, a figure that explains the statements in lines 81 – 104 may be helpful to understand these statements.
  5. Line 88: CID needs definition.
  6. Figure 1: More explanations are necessary. Right panels are enlargement of the portion of leftmost chromosome, but it is not obvious as ones in the tight panels look like whole chromosomes. It would be easier to see if the blue portion of chromosome is flanked by the red-marked portions at the both ends in enlarged views. Labels inside the figure are too small to see.
  7. Line190: I believe that the pole-facing edge corresponds to the “core” region (not the “peripheral” region in Dechat et al., 2004, equivalent to the “non-core” region in Haraguchi et al., 2008). The nuclear envelope reformation starts at the core region corresponding to the pole-facing edge. Therefore, the conclusion is correct, but the terms used seem to be mixed up. The “core” region proteins and “non-core” region proteins also seem to be mixed up. Those proteins are as follows: “core” region proteins include BAF, lamin A, emerin, LAP2α, and MAN1; “non-core” region proteins include lamin B, LBR, and Nups.
  8. Figure 2: Labels inside the figure are too small to see.
  9. Very trivial comments:

    Line 262: Section 3.2 should be 3.3 here.

    Line 6: The e-mail for W.S. continues to extra words “[email protected]”.

    Line 287: Do you really want to acknowledge for the Snickers issue?

Round 2

Reviewer 2 Report

I have no further comments. The authors still do not really address the yeast effects to my satisfaction, but I defer to the other reviewers.